# CSAW-M: An Ordinal Classification Dataset for Benchmarking Mammographic Masking of Cancer

**Moein Sorkhei**[*1, 2]  **Yue Liu**[*1, 2]  **Hossein Azizpour**[1]
**Edward Azavedo**[3,4]  **Karin Dembrower**[3,5]  **Dimitra Ntoula**[4]
**Athanasios Zouzos**[3,4]  **Fredrik Strand**[3,4]  **Kevin Smith**[1, 2]

[1]KTH Royal Institute of Technology, Stockholm, Sweden
[2]SciLifeLab, Stockholm, Sweden
[3]Karolinska Institutet, Stockholm, Sweden
[4]Karolinska University Hospital, Stockholm, Sweden
[5]Saint Göran Hospital, Stockholm, Sweden

## Abstract

Interval and large invasive breast cancers, which are associated with worse prognosis than other cancers, are usually detected at a late stage due to false negative assessments of screening mammograms. The missed screening-time detection is commonly caused by the tumor being obscured by its surrounding breast tissues, a phenomenon called masking. To study and benchmark mammographic masking of cancer, in this work we introduce CSAW-M, the largest public mammographic dataset, collected from over 10,000 individuals and annotated with potential masking. In contrast to the previous approaches which measure breast image density as a proxy, our dataset directly provides annotations of masking potential assessments from five specialists. We also trained deep learning models on CSAW-M to estimate the masking level and showed that the estimated masking is significantly more predictive of screening participants diagnosed with interval and large invasive cancers – without being explicitly trained for these tasks – than its breast density counterparts.

## 1 Introduction

Regular mammographic screening helps detect breast cancer at an early stage, and has been demonstrated to decrease mortality by around 30% [1]. However, 17-30% of breast cancers among screening participants are *interval cancers* – cancers detected clinically after a negative screening [2]. Interval cancers are often associated with a worse prognosis [3, 4]. So-called *true interval cancers* are characterized by rapid growth after a healthy mammogram, while *missed interval cancers* are the result of false-negative assessments of a mammogram, often because the lesion is obscured or *masked* by breast tissue.

Masking refers to the phenomenon in which a tumor is hidden by the surrounding breast tissue, causing the cancer to be difficult or even impossible to discern with regular mammography, as seen in Figure 2. Masking can also result in *large invasive cancers*[2] – a small cancer may be difficult to discern in certain images, allowing it to grow to a more lethal size. Masking is correlated with breast density, as it has been shown that cancer in dense breasts is more likely to be missed during screening [5, 6, 7]. Density can be subjectively assessed by radiologists via the *BI-RADS*

---

[*]Equal contribution

[2]We define large invasive cancers as those confirmed to have spread and be $\geq$ 2cm at time of diagnosis.

35th Conference on Neural Information Processing Systems (NeurIPS 2021) Track on Datasets and Benchmarks.

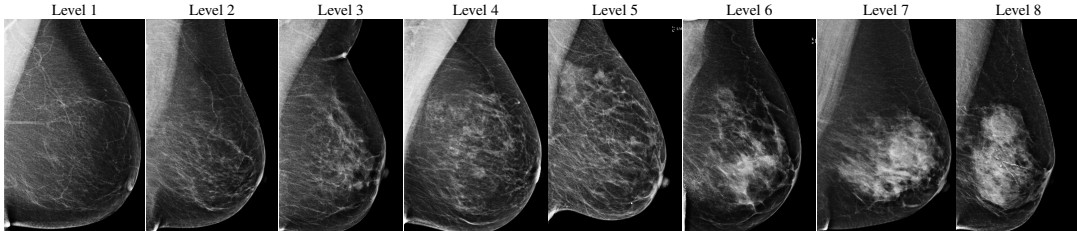

| Level 1 | Level 2 | Level 3 | Level 4 | Level 5 | Level 6 | Level 7 | Level 8 |

Figure 1: Masking potential, or the possibility for cancer to be obscured in a mammogram, from CSAW-M. From left to right, the most agreed-upon images among five expert annotators for level 1 (lowest masking potential, easist to assess) to level 8 (highest masking potential, hardest to assess).

*density* standard (ACR) [8, 9], or measured by automated tools such as Libra [10]. These density measurements, however, do not perfectly correlate with masking potential. Radiologists consider the distribution and pattern of tissue when assessing masking potential, and have called for automated methods to assess the masking effect [11]. Until now, the question of exactly how masking potential should be quantified remains an open one, although some subjective notion has been added to certain categories of the most recent edition of BI-RADS density [12].

The ability to assess masking potential is crucial because it can identify screening participants most likely to benefit from supplementary radiological methods, *e.g.* MRI. MRI is more sensitive than mammography, and has been proven to reveal tumors missed in regular mammographic screens [13]. Unfortunately, MRI is too costly and cumbersome to screen the whole population. The ability to predict high masking potential would allow clinics to identify screening participants most likely to benefit from MRI screening. These participants could be offered additional screening, potentially detecting more cancers as demonstrated in the DENSE trial [13]. Additionally, the ability to identify screening participants with low-masking potential – fatty breasts where tumors are obvious – would help hospitals more effectively allocate radiological expertise.

In this work, we introduce the **CSAW-M dataset** – a collection of over 10,000 mammographic images and associated masking assessments from experts. The assessments were graded by radiologists according to 8 levels of masking potential, as depicted in Figure 1, from easily assessed mammograms with low-masking potential (level 1) to difficult-to-assess examples with high-masking potential (level 8). This data can be used to train models capable of predicting masking potential from mammographic images in an ordinal classification setting.

The unique features of CSAW-M include:

1. It is the first dataset to directly address masking potential in mammography using expert assessments.

2. Aside from the masking potential assessments from five experts, CSAW-M also includes objective *clinical endpoints*, *i.e.* data on whether the screening participants developed interval or large invasive cancers.

3. CSAW-M is the largest public collection of mammograms, containing digital mammograms from over 10,000 screening participants, which can be repurposed for other tasks.

4. CSAW-M is distributed with a public test set for researchers to benchmark themselves. In addition, we defined a private test set which will not be distributed. An evaluation service hosted at SciLifeLab will allow researchers to submit Dockers containing their models for evaluation on the private test set, as a control against overfitting to the public test set.

In addition to these features, we provide a detailed analysis of expert agreement w.r.t. masking potential, and compare their performance to a baseline model we developed. We release the source code for our annotation tool, the implementation of baseline models and metric calculations, as well as the trained models[3]. Furthermore, the data contained in CSAW-M is relevant to the following research areas:

---

[3]Code available at: https://github.com/yueliukth/CSAW-M/.

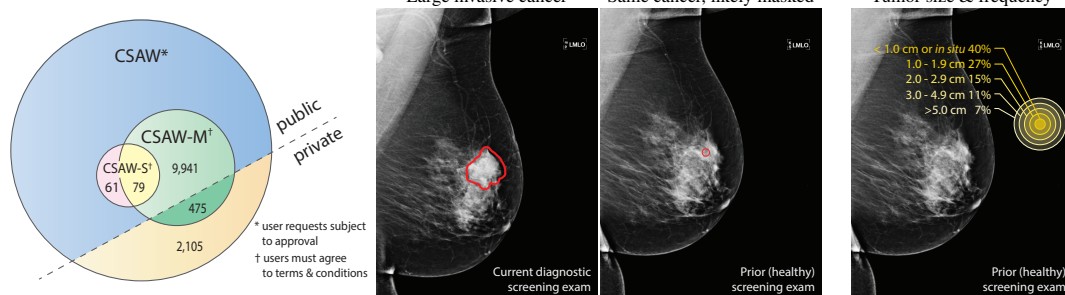

Figure 2: *(left)* The CSAW family of datasets with number of screening participants indicated. *(right)* From left-to-right, a mammogram of a large invasive cancer, the prior exam of the same cancer where the tumor was likely masked, and a visualization of tumor size & frequency at detection [17]. Note that, to maintain a pure notion of masking potential, CSAW-M images do not depict any identified cancers.

Table 1: Public mammography datasets.

| | Year | Origin | Participants | Images | Image modality | Masking metadata | Cancer metadata |
|---|---|---|---|---|---|---|---|
| MIAS [18] | 1994 | UK | 161 | 322 | Film | Non-ACR density | Cancer center & radius |
| LLNL [19] | 1995 | USA | 50 | 198 | Film | Non-ACR density & subtlety [5] | Cancer ROI, pixel-level |
| BancoWeb [20] | 2010 | Brazil | 320 | 1,473 | Film | Non-ACR density | ROI available in a few images |
| INBreast [21] | 2012 | Portugal | 115 | 410 | Digital | ACR density | Cancer ROI, pixel-level |
| BCDR [22] | 2012 | Portugal | 1,734 | 7,315 | Film & Digital | ACR density | Cancer ROI, pixel-level |
| CBIS-DDSM [23] | 2017 | USA | 1,566 | 3,103 | Film | ACR density & subtlety | Cancer ROI, pixel-level |
| CSAW-S [24] | 2020 | Sweden | 140 | 274 | Digital | - | Cancer ROI, pixel-level |
| CSAW-M | 2021 | Sweden | 10,020 | 10,020 | Digital | Explicit expert assessment | Interval/large cancers, image-level |

- **Ordinal classification/point-wise ranking**: labels in CSAW-M are ordinally related (ordered from 1 to 8). Few image datasets support the development and benchmarking of ordinal classification models, which is of value to the ML community.

- **Better pre-training**: recent works have shown that ImageNet [14] pre-training can be outperformed by pre-training on datasets of similar domain, which provides a better initialization [15]. CSAW-M can be valuable for pre-training models for tasks in a similar domain, *e.g.* cancer detection in mammograms.

- **Noisy labels and annotator agreement**: the masking potential labels in CSAW-M are subjective assessments. As such, the opinions from 5 expert radiologists we collected can be valuable for researchers investigating the effects of human noise and bias in the annotation process (and ways to mitigate these effects or use them for modelling aleatoric uncertainty).

CSAW-M is publicly available for non-commercial use [4]. It is hosted by the SciLifeLab Data Repository, a Swedish national infrastructure for sharing life science data. A *datasheet* [16] summarizing CSAW-M, along with detailed documentation, can be found in the Appendix M.

Although CSAW-M represents the largest public collection of mammographic images, a number of other mammography datasets exist. These datasets, summarized in Table 1, vary by modality, number of patients, demographics, and metadata provided. Most are focused on tumor detection, although some provide density measures along with the metadata. Existing public datasets are limited by the number of examples, the modality (scanned film is of inferior quality to digital mammograms), and the lack of explicit masking assessments. It is important to note that, unlike other datasets, CSAW-M does not contain images of cancers, so it is not directly useful for cancer detection.

---

[4]Dataset DOI: `10.17044/scilifelab.14687271`

[5]Subtlety is a subjective rating of difficulty in viewing the abnormality in the image, as defined in [19, 23], while masking potential discussed in this paper considers cancer-free mammograms.

Table 2: Summary of the CSAW-M dataset.

| | # images | Resolution | # interval / large invasive / total cancers | # composite endpoints | # controls | # masking annotations | Masking levels | Metadata | Publicly available? |
|---|---|---|---|---|---|---|---|---|---|
| Public train | 9,523 | 632×512 | 148 / 279 / 629 | 347 | 8,894 | 1 / image | 1-8 | Density, acquisition | Yes |
| Public test | 497 | 632×512 | 11 / 13 / 31 | 19 | 466 | 5 / image | 1-8 | Density, acquisition | Yes |
| Private test | 475 | 632×512 | 81 / 111 / 272 | 158 | 203 | 5 / image | 1-475 | Density, acquisition | No |

## 2 CSAW-M dataset creation

CSAW-M consists of screening mammograms along with metadata describing expert masking potential assessments, clinical endpoints, density measures, and image acquisition parameters. CSAW-M is part of an ecosystem of mammography datasets based on the CSAW population-based cohort [25], depicted in Figure 2. CSAW is a collection of millions of screening mammograms of screening participants aged 40 to 74 gathered from the three breast centers of the Stockholm region between 2008 and 2015. The CSAW *case-control* dataset, hereafter referred to as CSAW for brevity, is a subset of the full CSAW cohort containing all cancers, along with a random sampling of healthy screens from the Karolinska breast center. A portion of CSAW (2,580 screening participants) is designated as a private held-out test set, unavailable to the public, for controlled benchmarking of various tasks. CSAW-M is subset of CSAW, created here, to study masking. It is divided into a training set (9,523 examples), a public test set (497 examples), and a private test set corresponding to those in CSAW (475 examples). A summary of the dataset is provided in Table 2. CSAW-M partially overlaps with CSAW-S, a sister dataset focused on segmentation in mammograms [24]. Below, we describe the procedure followed to create CSAW-M depicted in Figure 3.

**Image selection.** Screening participants from CSAW were selected for inclusion to CSAW-M according to a flowchart found in Appendix A. As shown in Figure 3, starting from the CSAW population, we selected participants with mammographic screening exams from Karolinska University Hospital acquired with Hologic devices after the data was curated. From these sets of participants, we selected images as follows: the most recent mediolateral oblique (MLO) view of the breast was included, since MLO offers the best visualization of the breast [26]. If a selected participant had cancer, we selected the image of the contralateral breast (the one without cancer) to avoid contaminating the masking potential annotation task with actual tumors. Otherwise, the image was chosen with a random breast side. This resulted in screening participants fitting our selection criteria. To form our private test set, we finally sampled from the participants who correspond to the private split. From the participants belonging to the public split, we included all with cancer and sampled from the healthy population (as described below) to form our public data.

The images selected with our selection criteria above had a strong positive skew (light blue in Figure 3 and Figure 8 of the Appendix) in terms of percent breast density computed by Libra [10]. The most clinically interesting samples – very dense and very fatty breasts – belong to the under-represented tails of the distribution. We under-sampled the center of the distribution while keeping all samples of the tails, which resulted into a more uniform distribution (dark blue). Figure 3 shows an overview of the selection procedure.

**Image preprocessing.** The source images of our dataset are DICOM format files which are resized to $632 \times 512$ and saved with 16-bit PNG format as raw data of CSAW-M. Using the DICOM metadata, we perform a horizontal flip to make all breasts left-posed and rescale the intensity linearly into a proper DICOM window range. We locate the centroid of the breast and move it horizontally to the center of the image. Zero-padding is applied on the images in order to ensure uniform size among the images. The text in images (which includes the initials of the technician, breast laterality and view position) is removed by extracting the contour that is the closest to the top right corner of the image. Finally, the preprocessed images are saved as 8-bit PNGs. Further details are provided in Section B of the Appendix.

**Annotation procedure.** The goal of the annotation procedure was to label each image with expert assessments of masking potential. Masking was quantized into 8 bins, or *levels*, as depicted in Figure 1, for the public training and test sets. Images in the private test set are fully sorted according to masking. Individually sorted examples provide a more granular assessment, but at the cost of

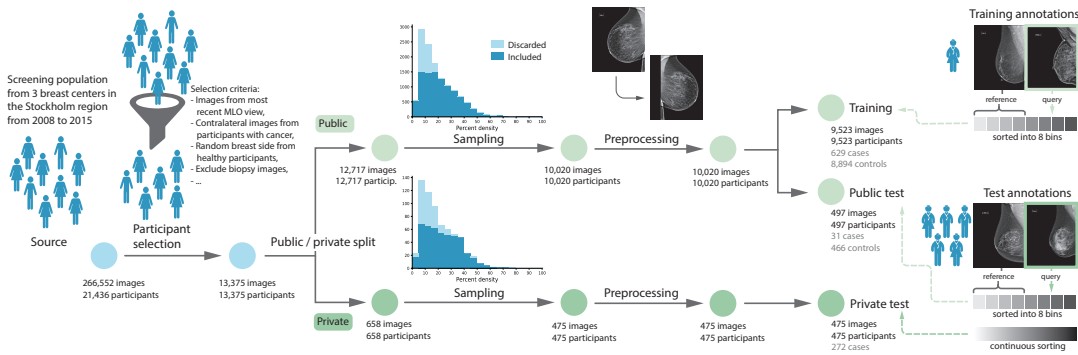

Figure 3: Overview of the creation of CSAW-M. From a screening population in Stockholm, participants were selected based on criteria described in Appendix A. The data was divided into public and private sets and then sampled to obtain a more uniform distribution of breast density. The images from the selected participants were preprocessed and annotated by experts – sorted using pairwise comparisons into 8 bins (public) or fully sorted (private). Please refer to the text for details.

increased annotation time. We opted for fine granularity on the private test set because *(1)* it allows for a more fine-grained assessment, and *(2)* it allowed us to identify robust initial bins[6] for the 8 masking levels in the public training/test sets. To represent the initial bins, we chose images personalized to each radiologist, but with highest agreement among the other experts. The benefit to this approach is that the starting point respects subjective assessments while at the same time choosing representative examples for each masking level.

Five experts contributed annotations to CSAW-M, each a licensed radiology specialist from the Stockholm region. Their experience in breast radiology ranged from 2 years to over 30 years. Note that the annotation of masking potential is a novel task for the radiologists. That is, it is not performed in their routine clinical practice, nor was it a part of their training. The training set contains one expert annotation per image, while the public and private test sets have five per image. We provide the individual assessments as well as a *ground truth* for the test set, computed as the median annotation. The median was chosen because *(1)* it is robust to outliers, *(2)* if there is a majority vote, median always selects it, and *(3)* it simplifies the process of discretizing masking levels.

The annotation procedure itself was based on a principle of *pairwise comparisons*. As depicted in Figure 3, an annotation software tool presented radiologists with a pair of images, a query $q$ and a reference $r$ (see Appendix C details). The radiologist was tasked with *deciding which image has the higher masking potential* – or, put another way – *which image is harder to be certain there is no tumor?* Based on the experts response *(query q, reference r, or "no difference")*, the query image was sorted relative to the reference image. Through a series of such comparisons, similar to a binary or ternary search, images were sorted either into 8 masking levels, or down to the individual images, see Figure 10. We use pairwise comparisons because they are more meaningful and repeatable for the experts than the arbitrary assignment of ordinal labels.

We chose a granularity of 8 masking levels for several reasons. Eight masking levels meant that, at most, 3 pairwise comparisons were necessary to sort each image. This appeared to be an acceptable compromise between granularity and annotation cost, as higher granularity appeared to limit the tendency of the experts to agree.

**Private test set.**    Creation of the private set started with a common *seed list* of 6 sorted mammograms selected by one of the experts. Each radiologist was given this list, and expanded it to a list of 500 individually sorted images through pairwise comparisons, via a strategy described below.

Given a query image $q$, we try to find a suitable position to place it so that the list remains sorted. This is accomplished by comparing $q$ with multiple images from the list, as shown in Figure 10 of Appendix C. To enforce consistency in the annotations, we devised a method inspired by the *ternary search* algorithm. Suppose we want to insert $q$ in the interval $[l, h]$. We compare $q$ against two *anchor* images $a_1$ and $a_2$ positioned at the $p_1 = (l + h)/3$ and $p_2 = 2 \cdot (l + h)/3$. The ternary comparison amounts to two consecutive pairwise comparisons presented to the expert, where the query image $q$

---

[6]We use "bin" and "level" interchangeably to denote collections of images with similar masking potential.

is compared against each anchor, $a_1$ and $a_2$ consecutively. The answers determine where $q$ is placed, and logical checks on the answers ensure the expert answered consistently.

Ternary comparisons ensure annotator consistency, but are costly. Moreover, as the depth of the search increases, expert self-consistency decreases. Hence, we used ternary search for the first two steps of each sorting, after which we employed a *binary search* based method. In the binary search, given the search interval $[l, h]$, the image at position $p = (l + h)/2$ would be selected as the *reference* image. Technical details of the ternary and binary search are provided in Appendix C.

**Training and public test sets.** Image-level sorting, as performed on the private test set, was too costly to apply to the 10,020 images set aside for public training and testing. Therefore, for the public data, experts sorted images into discrete masking levels, with 8 levels chosen for the reasons described above. To begin we first created a personalized list of 32 images, 4 per masking level, for each expert. To accomplish this, each expert's sorted private list was divided into 8 equal bins. The average rank was defined as: $\tilde{r}_i = \sum_{j=1}^{5} r_i^j / 5$ where $r_i^j$ denotes the rank assigned by radiologist $j$ for image $i$. Similarly, the rank delta of an image $i$ w.r.t. different annotators was defined as: $\delta_i^j = |\tilde{r}_i - r_i^j|$. Each expert received as a seed for annotating the public data, 32 images – the top 4 of their own personally sorted images per bin, with lowest $\delta_i^j$ (the *most agreed-on* images).

As before, our goal was to sort the images. But this time, the search interval was over masking levels instead of fully sorted images. Given a query image $q$ and 8 masking levels, the initial search interval would be [1-8]. For each step in the search, we first selected a *reference bin* in the middle of the search interval, from which we took a random *reference image*. The query image was then assessed against the reference image using a pairwise comparison. Since we defined an even number of masking levels, a situation can occur where the middle of a binary search interval would lie between two bins. Rounding would result in some bins being selected more often than others. To prevent that, we make sure that bins have an equal chance of being selected. For example, the initial search interval is [1-8], so we start by randomly selecting bin 4 or 5 as the reference bin. Following that, whenever the middle of the binary search falls between two bins, we round it down if it is above 4 and round it up if it is below 5. This simple modification allows for *symmetrical* moving out from the middle and allows bins at different steps of the binary search to have an equal chance of being shown as reference. See Appendix E for the masking level distributions of each expert resulting from the annotation process.

**Summary of the CSAW-M dataset.** CSAW-M public data consists of 10,020 mammography images at $632 \times 512$ resolution in 8-bit PNG format, and associated metadata as described in Table 2. The metadata includes the masking potential labels collected through the annotation process (including the computed ground truth); clinical endpoints *i.e.* cancer attributes including `cancer`, `interval` and `large invasive`; image acquisition attributes including `laterality`, intensity window `center` and `width`; and density attributes including `percent density`, and `dense area`. The annotations were equally distributed among the annotators, with approximately 2,500 assigned to each. A few images included in the annotation process are not included in the final dataset for various reasons, *e.g.* missing/declined annotations or problems with the image. In general, the experts were able to agree as to the masking potential of mammography images, although some tended to agree more closely than others, as indicated in Figure 5. See the discussion in Section 4.

## 3  Experiments

We conducted experiments using simple models to *(1)* empirically analyze CSAW-M, and *(2)* serve as baselines for future work. We evaluated performance of these models for prediction of masking potential, as well as for the indirect clinical tasks of correlating masking potential estimates with the two clinical endpoints of interval cancers and large invasive cancers.

**Baseline models.** We trained two baseline models to predict masking potential from mammography images. Both use ResNet-34 [27] as the backbone.

- *ResNet34 one-hot* – This model uses a standard approach for categorical classification, where each class is treated independently. Masking levels are predicted independently using standard softmax and cross-entropy loss to predict one-hot encodings.

- *ResNet34 multi-hot* – This model accounts for the ordinal relation between masking levels using *multi-hot* encodings [28]. Training a model with multi-hot encoding could be seen as multi-label classification, where the task of classifying an item into $K$ ordinal classes is

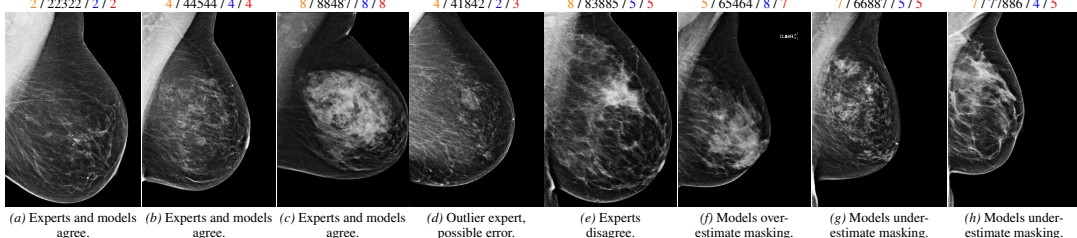

| 2 / 22322 / 2 / 2 | 4 / 44544 / 4 / 4 | 8 / 88487 / 8 / 8 | 4 / 41842 / 2 / 3 | 8 / 83885 / 5 / 5 | 5 / 65464 / 8 / 7 | 7 / 66887 / 5 / 5 | 7 / 77886 / 4 / 5 |

*(a)* Experts and models agree.  *(b)* Experts and models agree.  *(c)* Experts and models agree.  *(d)* Outlier expert, possible error.  *(e)* Experts disagree.  *(f)* Models over-estimate masking.  *(g)* Models under-estimate masking.  *(h)* Models under-estimate masking.

Figure 4: Agreement and disagreement amongst experts and models. Eight mammograms are shown, along with labels as follows: GT / experts 1-5 / one-hot / multi-hot.

equivalent to solving $K-1$ independent binary classifications, each class being a superset of the previous. For an ordinal classification problem with $K$ classes, the multi-hot encoding for datapoint $x_i$ with ordinal label $l_i \in \{1, ..., K\}$ is defined as $\{y_i^1, y_i^2, ..., y_i^{K-1}\}$ where for each $k \in \{1, ..., K-1\}$, $y_i^k = \mathbb{1}\{l_i > k\}$ where $\mathbb{1}\{.\}$ denotes the indicator function.

For both models, we used a batch size of 64 and trained on $632{\times}512$ images using the Adam optimizer [29]. Both are initialized with ImageNet-pretrained weights [14]. We used a learning rate of $1e{-}6$ and applied random horizontal and vertical flipping, random rotation of 10 degrees, and small random brightness and contrast jittering as data augmentation. We used 5-fold cross-validation to determine the stopping iteration, which we used in the final training run using all the training data.

**Task 1: Ordinal classification of masking potential.** The principal task of CSAW-M is to model the median expert opinion of masking potential level in the range [1-8]. Recall that the expert labels in CSAW-M are ordinally related, implying that a prediction confusing level 1 with level 8 is worse than one confusing level 1 with 2. We consider two metrics widely used to evaluate ordinal classification, *(1)* average mean absolute error *(AMAE)* which measures the average distance of predicted classes w.r.t. the true classes and is robust to class imbalance [30], and *(2)* Kendall's $\tau_b$ [31] which measures the correlation of two rankings based on the number of concordant and discordant pairs. Kendall's $\tau_b$ ranges from -1 (perfect inverse correlation) to 1 (perfect correlation), and 0 indicates no correlation.

The metrics described above consider performance over all masking levels. In addition, we consider model performance at identifying low- and high-masking mammograms. From a clinical perspective, these levels are most interesting because they represent cases the experts are most and least confident about. Participants with high-masking images can be *e.g.* offered additional screening. We consider the two lowest masking levels (1 and 2) together as *low-masking levels*, and the two highest ones (7 and 8) as *high-masking levels*. This choice was based on feedback from the experts. To assess how the models perform at identifying images in these *tail* masking levels, we use *F1-score* – a common metric to assess the performance in information retrieval.

**Task 2: Identification of interval and large invasive cancers.** A secondary task of high clinical relevance is to measure the correlation between predicted masking estimates and certain cancers. In particular, we consider how masking potential can predict *interval cancers* and *large invasive cancers* without being explicitly trained for these tasks. We measure performance using area under the ROC curve (AUC) for individual cancer types and for the *composite endpoint* (CEP) containing both types. We also calculate the *odds ratio* (OR), a measurement often used in clinical studies. The odds ratio is simply the ratio of the odds of an event occurring in one group to the odds of it occurring in another group. To compute it, model predictions are divided into groups $g_i$. Here, groups correspond to quartiles of the model predictions, so $i \in \{1, 2, 3, 4\}$. For screening participants in group $g_i$, the odds of having interval cancer is computed as $O_i = IC_i / \tilde{IC}_i$, where $IC_i$ is the number of participants from $g_i$ with interval cancer, and $\tilde{IC}_i$ is the number without. The odds ratio is then computed relative to a reference group, in this case the reference is the first quartile $g_1$, as $OR_i = O_i / O_1$. Note that $OR_1 = 1$ for all models. If a masking estimate is a good predictor of interval cancer it will show strong odds ratios in the top quartiles and exhibit monotonically increasing odds ratios.

We compare our baseline models with *dense area* and *percent density* computed using Libra [10]. This was done because density is known to be correlated with the clinical endpoints. For a fair comparison, it was necessary to convert the discrete masking predictions from our models to continuous values. We compute a continuous score as the weighted average of probabilities that an input belongs to each masking level. Refer to Appendix D for details.

Table 3: Comparison of expert and model performance on ordinal classification of masking potential for the public test set. Mean and standard deviation of 5 runs are reported for the models.

|  |  | Kendall's $\tau_b$ ↑ | AMAE ↓ | $F_1$ on level 1-2 ↑ | $F_1$ on level 7-8 ↑ |
|---|---|---|---|---|---|
| Experts | Expert 1 | 0.7232 | **0.6762** | 0.7940 | 0.6154 |
|  | Expert 2 | 0.7279 | 0.7167 | 0.7465 | **0.6316** |
|  | Expert 3 | 0.5450 | 1.0037 | 0.7363 | 0.5200 |
|  | Expert 4 | 0.5554 | 1.0390 | 0.5430 | 0.6242 |
|  | Expert 5 | 0.6342 | 1.0321 | 0.6885 | 0.5225 |
| Models | One-hot | 0.7126 ± 0.0083 | 0.8108 ± 0.0145 | 0.7855 ± 0.0136 | 0.5950 ± 0.0243 |
|  | Multi-hot | **0.7625 ± 0.0030** | 0.7086 ± 0.0142 | **0.8064 ± 0.0188** | 0.5571 ± 0.0320 |

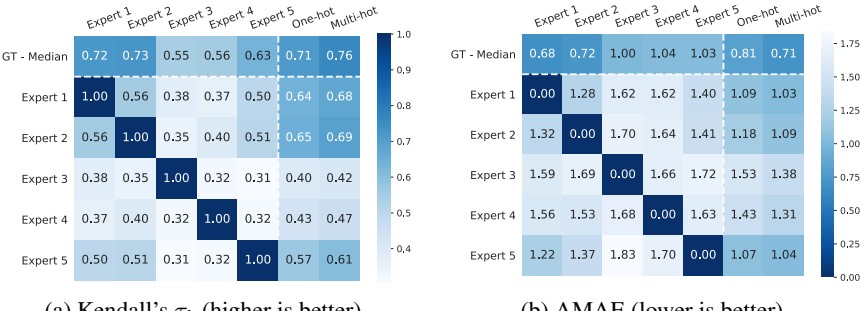

(a) Kendall's $\tau_b$ (higher is better)    (b) AMAE (lower is better)

Figure 5: Expert and model agreement on public test set. We perform five runs on our one-hot and multi-hot models and report the mean. See Appendix F for results on the private test set.

## 4 Results and discussion

**Expert agreement.** We begin by considering the question *how well do the experts agree w.r.t. masking potential?* This is an important question to consider, as the main task is to emulate the median expert assessment. Table 3 shows experts have an *AMAE* ranging from 0.68 to 1.04. This suggests that, on average, individual experts are almost ±1 masking levels distant from the ground truth – a reasonable level of agreement. A more nuanced picture of expert agreement is given in Figure 5. Here, agreement between each expert, as well as the ground truth, is measured by Kendall's $\tau_b$. As a rule-of-thumb Kendall's $\tau_b \geq 0.3$ indicates a strong association [7]. According to this rule, all experts have a strong association, although we can see that experts 1, 2, and 5 exhibit substantially higher agreement than experts 3 and 4. Interestingly, the experts who tended to agree more were also less experienced. Turning to the F1-scores in Table 3, it is clear that experts are in better agreement for low-masking cases than for high-masking cases. This suggests that high-masking potential is a generally less *agreeable* property than low-masking potential. Our findings on the public test set are mirrored in the private test set, provided in Appendix F. Examples of mammograms where experts agree and disagree are provided in Figure 4.

**Task 1: Ordinal classification of masking potential.** Results for *ResNet34 one-hot* and *ResNet34 multi-hot* (5 runs each) are provided along with the expert agreement in Table 3 and Figure 5. We can see that the model designed for ordinal classification, *ResNet34 multi-hot*, outperforms the standard one-hot model according to most metrics: Kendall's $\tau_b$, AMAE, and low-masking F1. The two metrics most sensitive to ordinal relations (Kendall's $\tau_b$ and *AMAE*) show a large gap between the two models, suggesting multi-hot encoding is more effective in solving our ordinal classification problem. The only metric where the one-hot model dominates is the high-masking F1-score, although both seem to struggle (as do the experts). This comes as something of a disappointment, as high-masking patients are the most interesting from a clinical perspective. On the other hand, both models performed excellently at identifying low-masking mammograms, and outperformed all the experts.

The models seem to be more correlated with Experts 1, 2, and 5, who agreed with each other more often (and the ground truth). Interestingly, our models were more correlated with each individual radiologist than any of their colleagues were (see the two rightmost columns in Figure 5). Please note that the cross-tabulation in Figure 5b is *asymmetric* because the number of mammograms

---

[7]We refer the reader here for an interpretation of Kendall's $\tau_b$.

Table 4: AUC on downstream clinical tasks public and private test sets combined.

| | AUC | | |
| | Interval cancer | Large invasive cancer | CEP |
| --- | --- | --- | --- |
| Percent density | 0.5947 | 0.5254 | 0.5678 |
| Dense area | 0.5901 | 0.5505 | 0.5839 |
| One-hot | $0.6321 \pm 0.0031$ | $0.5801 \pm 0.0013$ | $0.6100 \pm 0.0013$ |
| Multi-hot | $0.6331 \pm 0.0031$ | $0.5802 \pm 0.0019$ | $0.6117 \pm 0.0028$ |

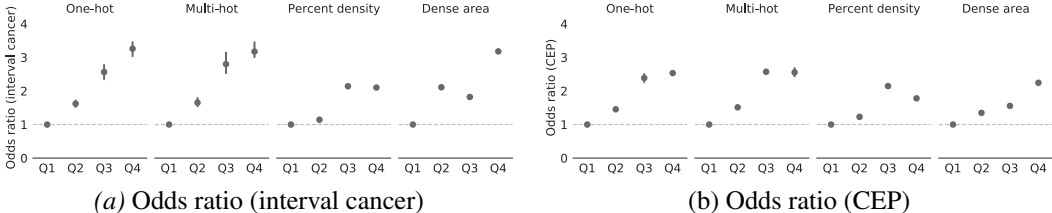

*(a)* Odds ratio (interval cancer)       (b) Odds ratio (CEP)

Figure 6: Odds ratio on clinical endpoints with public and private test sets combined.

placed into each masking level is different for each radiologist – *i.e.* the *AMAE* between two experts changes depending on which one is considered as the reference. In Figure 4, we provide several examples where the networks both agree and disagree with the experts. Figure 4*(g)* is an interesting case because the density is fairly low but experts rate it as high masking potential. Our models under-estimate the masking, suggesting they rely too heavily on general density cues.

**Task 2: Identification of interval and large invasive cancers.** We set out to investigate if estimates of masking potential are predictive of *(1)* interval cancer, *(2)* large invasive cancer, and *(3)* the composite endpoint (CEP), *i.e.* interval *or* large invasive cancer. There is reason to believe a correlation exists, because interval cancers or cancers that appear to grow fast can result from a misdiagnosed mammogram, which is more likely if masking potential is high. Due to the low number of cancers randomly sampled into the public test set, this part of the analysis is performed on the combined public and private test sets for increased statistical power. In Table 4 we compare our models against *percent density* and *dense area* at identifying these outcomes, as measured by AUC. Both models are stronger indicators than the density measures by a large margin, with *ResNet34 multi-hot* slightly outperforming *ResNet34 one-hot*. All methods perform stronger at predicting interval cancer than large invasive, which makes sense because the relation to masking is more likely to be causal – *e.g.* a missed diagnosis caused by masking.

In Figure 6 we plot the *odds ratio* for successive quartiles of the predictions from various models. Recall that good clinical predictors should exhibit monotonically increasing odds ratios, with strong odds ratios in the highest quartiles. If masking is related to interval cancer, then a high-masking prediction should have much higher odds to identify a cancer than a low-masking prediction. We can see that this is the case for both *ResNet34* models, with odds of finding a interval cancer 3.2 times higher in the top quartile than the first. In contrast, the density measures yield lower odds ratios and fail the monotonicity test, indicating they do a poor job of sorting cancer risk. The odds ratios for large invasive cancers, given in the Appendix G, are less promising. This trend is also evident in the AUC results from Table 4, suggesting that identifying screening participants confirmed with large invasive cancer is more challenging than interval cancer. The composite endpoint, which combines both cancers, reflects this in the AUC and odds ratios.

## 5 Conclusions

There is a strong clinical interest in predicting interval and large invasive cancers, as they indicate a failure of the screening infrastructure and may lead to poor prognoses. We have shown that deep learning models trained on CSAW-M can identify these cancers significantly better than density measures. This suggests that the expert masking potential information provided by CSAW-M has high clinical value, aside from its value as an ordinal classification benchmark and the other merits listed in the introduction. Our baseline models succeeded at modeling masking potential, agreeing with the ground truth better than any individual expert. However, agreement among experts and our models was much better for low-masking than for high-masking. We note that our baselines are very

simple models that were not explicitly trained to learn the clinical outcomes. As such, there is much potential for improvement.

**Limitations and broader impact.** CSAW-M is primarily intended for the development of automated systems to estimate masking of mammograms. Such a system can be used in the screening process to pre-emptively detect interval and large invasive cancers, as we are attempting to show in an ongoing clinical trial, ScreenTrust MRI[8]. Furthermore, since CSAW-M is the largest public mammography dataset, it can enable more fruitful research and development for different applications that involve automatic analysis of mammograms, for instance, through unsupervised learning. Such automated systems can help with the world-wide shortage of specialists. On the other hand, a general concern when releasing a human cohort dataset is malicious use of the released data to re-identify the individuals. Therefore, we have taken measures to mitigate this issue: (1) we removed all individual identifiers from the data, (2) we down-sampled the mammograms, (3) we removed all unnecessary acquisition attributes –DICOM headers–, (4) we simplified the continuous tumor size attribute to a binary outcome, and (5) we anticipated a gated release mechanism to approve users based on their information and project goals before granting access to the data. It should be noted, however, that while these efforts make re-identification extremely unlikely, it does not provide a theoretical guarantee. We acknowledge several limitations and biases present in CSAW-M. For example, there are no visible tumors in CSAW-M by design. This limits its usefulness for tumor detection. Also, the training set is limited to a single annotation per image, as a compromise between annotation cost and number of examples. This affects the training data more than the test data, as the ground truth is the median of 5 expert opinions. Biases are introduced from a number of sources, *e.g.* the training and background of the experts. We should also be aware of biases inherent in the collection process, as it was collected from a certain population, period, and region, using certain imaging equipment (*e.g.* we selected more screening participants with cancers than are present in the general population). To alleviate this, the *known* specifics of the population and our selection pipeline are thoroughly described in the paper. Nevertheless, clinical studies are crucially required before deploying models in any clinical processes.

**Acknowledgements.** This work was supported by the Regional Cancer Center Stockholm-Gotland as source for the clinical data and partly financed by MedTechLabs https://www.medtechlabs.se/, the Swedish Research Council (VR) 2017-04609, and Region Stockholm HMT 20200958. We thank Christos Matsoukas and Johan Fredin Haslum for an early review of this work.

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
