# OpenReview forum: "CSAW-M: An Ordinal Classification Dataset for Benchmarking Mammographic Masking of Cancer"
_NeurIPS.cc/2021/Track/Datasets_and_Benchmarks/Round1 — NeurIPS 2021 Datasets and Benchmarks Track (Round 1)_

### Official Review · Reviewer_qQn9 · 2021-06-24
**New mammographic dataset**

**Rating:** 8
**Confidence:** 4
**Clarity:** Yes, very well written.  Good job!

**Strengths:**

This proposition has many strengths:

* First and foremost, it is VERY large.  With 10,000+ patients, this dataset is the largest of its kind and thus has the potential of having a bold impact on the community.
* Annotations from multiple experts give a lot of credibility to the groundtruth.  It also provides inter-expert variability intervals.
* The dataset is well balanced and the images were selected via a sound evaluation protocol summarized by a flowchart provided in the appendix.
* source code for the annotation tool and baseline models have been rendered public
*  Considering that breast cancer is one of the most prominent cancer but whose survival rate is high when diagnosed early, it is easy to understand that better masking prediction will lead to more accurate early diagnostics and thus better survival rates.
*  The manual annotation procedure is thorough and explained in great details.







**Weaknesses:**

In my view, this work has two main limitations:

* that dataset is not really opened as it can only be used by non-profit organizations.  This drastically reduces its future impact on *real* clinical applications.  As a promoter of open data my self, it is of critical importance that source code (like Linux, python, tensorflow, etc) as well as datasets (like ADNI, PPMI, etc) be open to the entire community and, in particular, to those that bring solutions to the clinical world (namely private companies).

* I haven't been able to appreciate the online system the authors claim to have.  The dataset is hosted on a Swedish hosting system (SciLifeLab) that a priori only Swedish University members  can access.  I sent an email for a login but haven't got any feedbacks...  Their login access policy seams to be yet another effort to draw a line between those who can access the data and those who cannot.  I wish a different system like Kaggle or grand-challenge.org had been used.

**Additional Feedback:**

It is overall a good work that deserves to be published.

**Correctness:**

As mentioned before, the dataset is well balanced and the images were selected in a systematic way (details in the appendix).  Same with the manual annotation procedure which the authors have explained in great details.  Also, the fact that the images are in png format (instead of DICOM for example) makes it very easy to process.


**Documentation:**

The dataset is hosted on a system called SciLifeLab that I was not aware of before reviewing this paper.  It seams a well managed system, but i haven't been able to log onto it and appreciate it for myself.

Also, as mentioned before, I do not like that the license excludes for-profit organizations.

**Ethics:**

This project seams to have been approved by the Regional Ethical Review Board in Stockholm and the images are in png format (thus void of any patient meta-data).  I thus see no ethical issue regarding this project.

**Relation To Prior Work:**

In my opinion, Table 1 says it all:  the proposed dataset is between 1 and 2 orders of magnitude larger than what has been publicly released before.   CSAW-M is thus an important step forward.

**Summary And Contributions:**

This paper presents a new mammographic dataset called CSAW-M which happens to be a subset of CSAW, a large mammogram dataset.   CSAW-M contains data from a whopping 10,000+ individuals aged between 40 and 74 together with masks manually outlined  by 5 specialists.  As mentioned early in the introduction, masking is a phenomenon in which a tumor is hidden by surrounding [dense] tissues and thus more likely to be missed.  Images were also graded according to 8 level of masking.   The dataset is divided in 3 subsets, namely a public train, a public test and a private test dataset.  The data is stored on an online repo called "SciLifeLab".

The authors also trained 2 generic ResNet models and report results within the inter-expert variability (especially for the ordinal classification task).

---

> ### Author Response · Authors · 2021-07-13
> **Response to Reviewer qQn9**
>
> First, we would like to thank you for your helpful review and feedback on our submission. Below we try to address the points in your review.
>
> 1. Regarding your point about making the dataset public to the entire community: we acknowledge that the ideal scenario is to make data public to everyone, including private companies. However, preserving privacy when releasing medical images is of critical importance. We must act conservatively since it is difficult, if not impossible, to recall data if a privacy leak has been identified -- and we are personally responsible for the release of the data. While a dataset of this scale is of immense value to academic research, private companies engaged in this line of research typically have agreements with hospitals that give them access to orders of magnitude more data. In this sense, CSAW-M is of more value to academics, who typically do not have such agreements, than it is to private companies.
> \
> \
> Furthermore, the datasets you refer to, ADNI and PPMI, are in fact more restrictive in their accessibility and terms of use than CSAW-M. ADNI is restricted to non-commercial use and requires a more restrictive Data Use Agreement and an application including “the investigator’s institutional affiliation and the proposed uses of the ADNI data”. (please refer to: http://adni.loni.usc.edu/data-samples/access-data/#access_data). PPMI requires an online application with a restrictive Data Use Agreement where users request access to data “for the purpose of scientific investigation, teaching, or the planning of clinical research studies”, and comply with PPMI Publications Policy (please refer to https://www.ppmi-info.org/access-data-specimens/download-data/).
>
>
> 2. Regarding your point about the online system which hosts CSAW-M: we mistakenly included a wrong link in our manuscript. We pointed readers to  https://www.scilifelab.se/data/repository/ when we should have linked to https://scilifelab.figshare.com/. The link we sent was instructions for SciLifeLab researchers to upload their data. We encourage you to navigate to the correct link and view a few of the datasets and see how the interface works. Logins are not necessary (only for uploading data). In most cases, data can be downloaded with a single click.
> \
> \
> To clarify about the hosting system: the SciLifeLab Data Repository is a data repository hosted by SciLifeLab (the Swedish National Initiative in molecular biosciences https://www.scilifelab.se/) for researchers to upload and share data. Currently, this service relies on Figshare (https://figshare.com/) and is maintained by the SciLifeLab Data Center. Once the data is shared, everyone can view the dataset webpage and use the request access functionality. This means every person, regardless of being a Swedish university member, will be able to view the dataset webpage and use the data request functionality.

---

> > ### Comment · Reviewer_qQn9 · 2021-07-13
> > **Response**
> >
> > Unless I misread the answer of the authors, I do not see the link between privacy protection and not allowing private companies to use the data.  If there are any privacy issues with the data, then it shouldn't be released at all.  Also, the statement " private companies engaged in this line of research typically have agreements with hospitals that give them access to orders of magnitude more data" is a bold overstatement that does not fit to the reality of a majority of private companies.  For the vast majority of companies, prior to sign any agreement with an hospital, or to launch a clinical study, or to get involved with the development of a new product, one always start with proof of concepts typically obtained on public data that people can rely on (and I know what I am talking about, I have been doing this for years).
> >
> > As for ADNI, it can be used by private companies, it just cannot be sold, redistributed or included into a product.  With a simple google search, one can easily find publications (even patents!) written by private companies reporting ADNI results.
> >
> > That said, I agree with the authors that ADNI and PPMI have some restrictions, but overall the use of these datasets by public and private research groups have indisputably promoted science and the development of ground breaking technologies.

---

> > > ### Author Response · Authors · 2021-07-14
> > > **Response to Reviewer qQn9**
> > >
> > > Thank you for your response. As a point of clarification, we would like to mention that the license we have chosen, CC BY-NC-ND 4.0 (https://creativecommons.org/licenses/by-nc-nd/4.0/), prohibits uses that are “primarily intended for or directed toward commercial advantage or monetary compensation”, however, it does not prevent for-profit organizations (e.g. private companies) from using the data for research purposes, as the official CC instructions state [1].
> > >
> > > Our choice of the license was guided by the permission granted from the Ethical Review Board. The need for informed consent from the participants was waived by the Ethical Board under the explicit condition that the data would be shared with other researchers for research purposes. As such, researchers at private companies can also use the data, as long as the intended use is not primarily commercial (e.g. for running a clinical study). We believe this is in line with the point you mentioned about ADNI.
> > >
> > > References:
> > >
> > > [1] https://creativecommons.org/faq/#does-my-use-violate-the-noncommercial-clause-of-the-licenses

---

> > > > ### Comment · Reviewer_qQn9 · 2021-07-20
> > > > **Final review - '(modified: 20 Jul 2021)'.**
> > > >
> > > > Thank you for your answer.  I maintain my review:
> > > >
> > > > Rating: 8: Top 50% of accepted papers, clear accept

---

### Official Review · Reviewer_Zxbf · 2021-06-25
**Review of "CSAW-M: An Ordinal Classification Dataset for Benchmarking Mammographic Masking of Cancer"**

**Rating:** 8
**Confidence:** 4
**Clarity:** Yes - I appreciate the authors' clear…

**Strengths:**

I find the data to be well-organized with easily understandable annotations and an abundance of metadata. Overall, this dataset is a valuable resource that could potentially establish new lines of research, both in studying masking as a risk factor for mammography-specific applications, and in developing ordinal methods in machine learning. The manuscript is well-written, and explains their labeling procedure in detail.

**Weaknesses:**

One weakness is the limitations that the authors had with labeling the images: the training dataset images are each only annotated by one expert, as opposed to the test datasets that are a median of five experts. However, this is an understandable limitation given the nature of this dataset: labels from medical experts are just more expensive and difficult to procure than tasks that can be conducted by laypeople. I appreciate the authors outlining their logic in how they distributed their resources in light of these limitations, and it is evident that extensive thought has gone into making the best of a limited situation - I doubt many other medical image datasets of similar scale would be more thoroughly labeled than this one.

The main problem that I noticed with this dataset comes from browsing the data directly:

1. I noticed that a substantial portion of the images contain text at the top corner, presumably from the imaging instrument. I'm concerned that this text could be a confounder especially for deep learning models that may learn to memorize this text as a shortcut over actually examining the imaged breast. Could the authors provide a better explanation of what this text is, and if they expect any correlation with the text and masking or cancer? Does the text provide any potential of leakage of information or shortcuts in predicting the test datasets? Are there any best practices for preprocessing out this text, if necessary?

I think this is an issue that could potentially reduce the utility of this dataset for machine learning purposes, so I have given the paper a weak accept for now, but I am happy to improve my score if the authors can satisfactorily address this point.

I also have some additional thoughts on how the characterization of the dataset could be improved (but are non-essential), especially as it could better describe the exact nature of the ordinal problem here, that I will detail in the "Feedback to Authors" section.

**Additional Feedback:**

1. It looks like about half of the experts attempted to bin the all images evenly, while others have a more normal distribution to their labeling. Since there is still a heavy tail in the distribution of images even when undersampling their data, I expect the way this would interact with the classes is that for the experts that binned images evenly, you'd see a higher range of variability in the images of label 8, than in the images of label 3, for example. I'm less clear about what the intra-class distribution of variability would look like with the median labels: could the authors directly test this (e.g. one way I can think of is by looking at how percent identity distributes across each of the 8 ordinal labels, understanding this is an imperfect proxy)? I feel like this would be important to characterizing the exact nature of the ordinal challenge here: my feeling is that with the even binning strategy, there may be more overlap or less of a gap between most images of label 3 versus label 4, where there'd be a bigger difference between 7 and 8. (If the authors want to go the bonus mile, I would like to see how this interacts with the metrics for agreement between reviewers in Table 3 - are there certain pairs of adjacent classes where the disagreement is concentrated, or is it evenly distributed? And do the models mirror this confusion? The F1-scores provide some insight into this, but not directly.)

2. Could the authors clarify if identifying as a woman was an inclusion criterion for their images or for screening at the hospital site? If not, I would suggest that the authors use more accurate and inclusive nomenclature than "women" in their manuscript (e.g. "participants", "patients", or "people at risk for breast cancer"), given that other people including trans men and nonbinary people also receive mammograms.


**Correctness:**

Yes, the dataset is constructed in a sound way. As I discussed in the weaknesses, there are resource limitations in the collection of labels like these; the authors have proposed, in my opinion, sound principles in soliciting standardized labels from medical experts in light of these limitations.

**Documentation:**

Yes - the authors are collaborating with a data repository host with a solid plain for hosting. A clear creative commons license is associated with the data.

**Ethics:**

Typically, these medical datasets can have some privacy and re-identification issues associated with them, so I'm a little curious about why CSAW is gated by a dataset request mechanism, but CSAW-M only requires users consenting to terms and conditions. However, I do see this work has been approved by an ethics board, and de-identification procedures have been followed, so I assume diligence has been done here. I think given the biased nature of this dataset, especially since it comes from a single population, there are some external risks with future work that may attempt to use this dataset for medical claims as opposed to just methods building, but the authors have discussed this, and I see this as out of their control.

**Relation To Prior Work:**

Yes - similar mammography datasets are cited, and the authors make a clear distinction between their work and these datasets.

**Summary And Contributions:**

Here, the authors provide a new dataset of mammographic images, which focuses on masking potential as annotated by specialists. Their dataset provides several contributions over previous public mammography datasets: in addition to containing 1-2 orders of magnitude more images, they also focus on providing gold-standard masking annotations over pixel-level cancer ROIs.

---

> ### Author Response · Authors · 2021-07-13
> **Response to Reviewer Zxbf - Part 2**
>
>
> 4. Regarding Additional Feedback 1, concerning variability between adjacent masking levels: our revised manuscript contains two new sections, Appendix H and Appendix I along with Figure 13 and Figure 14, which address your questions.
> \
> \
> In Figure 13, violin plots show the distribution of percent density as a proxy, as per your suggestion, for each masking level and grouped by expert. As you hypothesized, experts  1&2, whose masking-level distribution was nearly uniform, showed high variability for high masking levels. Similarly, the change in the median of the violin plots for these experts increases for high masking levels.
> \
> \
> To “go the extra mile” and check how this trend relates to annotator agreements, we have added Figure 14, which is discussed in Appendix I. Here, we divided masking levels into 4 non-overlapping groups: (1, 2), (3, 4), (5, 6), and (7, 8). For each of these groups, we individually considered each expert as the reference (and GT-median), and evaluated how other experts/models agree with respect to the selected reference in terms of the *AMAE* measure. For example, in the top row of Figure 14a we consider the GT-median as reference, and compute the *AMAE* of the other experts. A low value indicates they agree with the GT-median in discriminating masking level 1 from masking level 2. We repeated this process for the other groups to fill in the remaining rows.
> \
> \
> From Figure 14, we can observe that, as a general trend, disagreement seems to be concentrated at high masking levels (7,8). The experts tended to agree best when discriminating between middle masking levels (3,4) and (5,6). Performance for the low masking levels (1,2) was interesting, as certain experts tended to agree with each other well (experts 1&2) while others disagreed strongly. The GT-median and the neural networks tended to reflect these trends, but generally agreed better than any individual expert.
>
>
> 5. Regarding your feedback about the use of the term “women”: more people than ever identify as non-binary, but Swedish legislation has yet to adapt to this fact. In Sweden, there are two legal genders, male and female, assigned at birth. A medical assessment is mandatory to change the gender assigned at birth. The personal identity number and passport reflect this legal status. Individuals with female personal identity numbers are invited for mammographic screening. They are legally women, but some may self-identify as a trans man or non-binary.
> \
> \
> Accordingly, we have adjusted the text throughout to use the term “screening participants” or “participants”, as suggested.

---

> ### Author Response · Authors · 2021-07-13
> **Response to Reviewer Zxbf - Part 1**
>
> We thank you for your insightful, detailed review and valuable feedback to our work. We have addressed your concerns about the confounding text that led you to lower your rating to “weak accept”, so we hope you will reconsider your rating accordingly. Regarding your other comments, find our point-by-point response below:
>
> 1. Regarding the training set containing one expert annotation per image: this concern was echoed by Reviewer Y1Wu -- we ask you to refer to our full response above. But, to briefly summarize, any data collection process will have some limited annotation budget. Given a choice between adding *3-5* times the number of examples or having more *3-5* annotators per training image, we believe the community benefits more by adding new examples as it is well-known that deep learning performance scales as data increases.
>
>
> 2. Here we address your question about the text that appears in some images. We hope this answers your concern that motivated your rating of “weak accept” and you reconsider the final rating.
> \
> \
> First, an explanation about what the text contains. The text includes 3 parts : a) the initials of the technician who adjusted the breast position and checks for image quality; b) breast side (L for left and R for right); and c) view position of the breast. In CSAW-M all breasts are MLO: mediolateral oblique, a standard mammographic view. As we stated in the paper (L107-110), we included a random breast side for healthy screening participants, and for participants with cancer we selected the contralateral side (the side without cancer). The side and view text cannot leak information about potential masking or cancer. The technician’s initials, although unlikely, DOES represent a potential confounder (e.g. if the hospital routes particularly hard cases to a certain technician).
> \
> \
> Therefore, we have updated the dataset and added a preprocessing step that removes the text from all the images. The manuscript has been updated to reflect this (except for a few images in the figures that still contain the text, which requires more time), see in Section 2 (L119-127). The new, text-free version of the dataset can be accessed using the same link as before, provided with the submission.
>
>
> 3. Regarding your concern about the apparent discrepancy between access procedures for CSAW and CSAW-M: currently, the full CSAW dataset can only be accessed by directly communicating with the authors, requesting they copy the data to a disk, and ship it. However, it is planned within a year, as part of an initiative among Swedish hospitals, that the full CSAW data will be made available under the same conditions as CSAW-M, with a data hosting plan and similar terms and conditions as defined in this work.

---

> > ### Comment · Reviewer_Zxbf · 2021-07-13
> > **Response to authors**
> >
> > Thank you to the authors for their extensive and thoughtful response. I appreciate that re-doing the entire dataset to remove the text is not a trivial feat especially given the short notice of the rebuttal period. The authors have addressed this concern, and they've also gone above and beyond in addressing my other non-essential feedback points. I will be raising my score during the discussion period to reflect this.
> >
> > I understand if this is not ready until the camera-ready, given the short period of the reviews, but have the authors had an opportunity to update their benchmarks with the new dataset as well? It is not essential for me to see this before the camera-ready, as it will likely only be minor changes to the benchmarks and the dataset is the main point of the paper, but I'm just personally curious if removing the text has had any changes on the performance of the models.

---

> > > ### Author Response · Authors · 2021-07-14
> > > **Response to Reviewer Zxbf**
> > >
> > > Thanks for your response and for considering raising the score!
> > > Yes, we are in the process of re-training the models and we will include the updated results in the camera-ready version.

---

> ### Comment · Reviewer_Zxbf · 2021-07-17
> **Updated score**
>
> I have updated my score in response to the discussion below.

---

### Official Review · Reviewer_Y1Wu · 2021-07-02
**Comments from reviewer (updated 19 Jul.)**

**Rating:** 7
**Confidence:** 3
**Clarity:** The paper is well written and easy to…

**Strengths:**

+ The scale of the proposed dataset is very large, compared to prior datasets, which could be beneficial for the research community.

+ The annotation (8 levels of masking potential) of the proposed dataset is novel and has real clinical value. The ordinal labels could also be used beyond the studied mammographic problem, e.g. benchmarking other ordinal classification models.

+ The data collection part of the dataset is well presented with sufficient details.

+ In addition to the proposed dataset, the authors also perform evaluations with baseline models on this dataset. The evaluation validates the strength of the proposed dataset with its novel annotation and also provides baseline models for following-up research.

+ The dataset is publically available for non-commercial use.

+ The proposed dataset has potential positive ethical and social implications, e.g. help with the automatic analysis of mammograms and research in breast cancer.

**Weaknesses:**

- It is a bit unclear how the dataset is hosted at the SciLifeLab. Besides, the mentioned "controlled, less biased setting" (L58) lacks details. It would also be better to provide more details of the "planned service" (L59).

- Although the collected dataset is quite large, it was collected only from one region (three centers of the Stockholm region). This could possibly introduce bias and there should have been a discussion on this.

- The collected data is from the age range of 40-74 (L90), but it is unclear how does this compare to existing datasets. It would be better to summarize and compare this as well in Table 1.

- Although more details are provided in the Appendix, the main paper lacks sufficient details (e.g. the preprocessing part) of how to construct the dataset.

- The images in the training set are only one expert annotation per image and the distribution of the expert experience for the whole training set is not provided. I know this is a large-scale dataset and it is costing to have multiple experts annotate each image, but only one expert annotation per image would inevitably introduce bias and affected by the experience of the expert. It would be much better if 1-2 more experts could be included for the annotation of each image. Or otherwise, an annotation distribution (analysis) in terms of the expert experience should be provided.

- In the test set, the "ground truth" is computed as the "median of the annotation". This is a bit inappropriate, especially considering the experience of the experts ranges from 2 years to 30 years.

- It is unclear why the masking level is defined to be eight and why has to be such an even number. The current description (L142-145) is a bit unconvincing.

- It is not mentioned how the proposed dataset will be used ethically and responsibly.

**Additional Feedback:**

* It would be better to move some content regarding the data processing and annotation analysis from the appendix to the main paper if possible, or clearly mention in the main paper what information is provided in the appendix.

**Correctness:**

The proposed dataset is generally constructed in a sound way. My only concern is the number of experts used to annotate each image and the definition of the masking level.

**Documentation:**

There is sufficient detail on the data collection and organization, availability, but lacks in the maintenance and ethical and responsible use parts. A URL is provided for the dataset with a hosting licensing documents. The maintenance plan is a bit not clear.

**Ethics:**

The collected dataset could possibly biased to people from a specific region, as mentioned above and in Table 1.

**Relation To Prior Work:**

The authors presented a discussion on how this work differs from previous contributions, though more details could be provided as mentioned above, e.g. the comparison on the age range and other metrics.

**Summary And Contributions:**

In this paper, the authors collect and present a new ordinal classification dataset for benchmarking mammographic masking of cancer. The dataset (named CSAW-M) is collected from 10,020 patients from Sweden and is the current largest public mammographic dataset with potential masking annotation. The authors also trained baseline deep learning models for evaluation and showed that the estimated masking is much more productive of women diagnosed with interval and large invasive cancers, than its breast density counterparts. The main contribution of this paper is the proposed dataset. Considering its large scale and the novelty in the annotation and its clinical value, it would benefit the research in the community.

I have read the rebuttal from the authors and for more details please see my response below.

---

> ### Author Response · Authors · 2021-07-13
> **Response to Reviewer Y1Wu - Part 2**
>
> 6. Regarding the use of median as the ground-truth for the test sets: we considered several options to define the ground-truth, but ultimately decided that median was the most appropriate choice. While it is true that the experts have different amounts of experience, it would be a mistake to weigh the annotations directly according to years of experience. A better approach would be to learn how to weigh the expert opinions using a method such as STAPLE or one of its derivative methods: COLLATE, SIMPLE, etc. However, an empirical study conducted in 2016 (Reference 3 below) found that constructing the ground truth by simply removing outliers/consensus voting gives similar performance to more “intelligent” approaches on well-behaved data, and is more stable when annotations have high variance (which is the case here). Furthermore, as we had only 497 test images to work with, simply taking the median makes more sense. Training and calibrating a learning-based model on this amount of data is problematic. Choosing the mean instead of the median is another alternative, but it tends to push the distribution further to the center -- an undesirable property as the dataset already naturally tends towards the center (which is the least interesting from a clinical perspective), and this would work against steps taken in the sampling stage to mitigate this.
>
>
> 7. Regarding the choice of 8 masking levels: defining the number of masking levels was not a straightforward decision. Much thought and discussion went into our choice. To clarify: the number of masking levels does not need to be an even number, and our text should not imply this -- please point us to any place where this seems to be unclear. We had three main considerations when choosing the number of levels at the start of the annotation process: 1) it should have appropriate granularity -- i.e. experts should feel that they can distinguish between adjacent levels, but the levels should not be so coarse that they are not useful, 2) it should be efficient in terms of pairwise comparisons -- annotations should not be wasted, 3) it should be easily divisible in case it became necessary to reduce the number of levels (e.g. go from 12 to 6 levels) -- a priori,  we did not know if expert agreement would be acceptable at the chosen granularity.
> \
> \
> Given these criteria, we arrived at the choice of 8 masking levels. We considered several candidates: 4, 8, 12, 24. The experts gave feedback that 12 (and 24) levels were too fine -- they could not decide between adjacent levels. Furthermore, the number of pairwise comparisons pushed the annotation budget. 4 levels was deemed too coarse to be useful for the downstream tasks.
>
>
> 8. Regarding the ethical and responsible use of CSAW-M: As the data contains personal images, the main concern with CSAW-M is privacy. We took several steps to protect privacy through our terms and conditions of use. As mentioned in Appendix K, we have chosen the CC BY-NC-ND 4.0 license (https://creativecommons.org/licenses/by-nc-nd/4.0/) for our dataset. This restricts modifications of the dataset and limits it to non-commercial use. In addition, we have defined terms that users must to agree to before accessing the data, including:
>
>     - Users will not attempt to re-identify the subjects or use the data for any illegal purposes
>     - In the unlikely case of re-identification of an individual by a user, the user is asked to immediately let us know
>     - Users will not share the download link to the data to others
>     - Upon our request, the user must immediately delete the data (e.g. in case of a privacy breach)
>
>   The terms and conditions give guidance on how the data should be ethically and responsibly (more details can be seen on the dataset website). We believe users who agree to our terms will keep ethics in mind when using the data.
>
>
> References:
>
> [3] Lampert, Thomas A., André Stumpf, and Pierre Gançarski. "An empirical study into annotator agreement, ground truth estimation, and algorithm evaluation." IEEE Transactions on Image Processing 25.6 (2016): 2557-2572.

---

> > ### Comment · Reviewer_Y1Wu · 2021-07-14
> > **Response to authors**
> >
> > Thanks to the authors for their detailed response, which addressed most of my concerns. I appreciate that it is costly for data collection, especially for medical data, and was not trying to push the authors to increase the number of expert annotations per image, though more would be better. It is true that usually "deep learning performance scales as data increases", but it also depends on the quality of the annotation if the learning process would be supervised. The community would benefit from more examples, but I would still recommend increase the quality of annotation and eliminate the bias as much as possible, especially for medical data. On the other hand, assigning an equal number of images (~2500) to each annotator would surely introduce bias, especially considering the large diversity of experience for the annotators. It would be better if the authors could include the discussion of this in the revised paper.
> >
> > Regarding the definition of the ground truth, though the authors claim median to be the best choice among other alternative solutions and gave a reference study, I would recommend the authors to include a (brief) comparison to other alternative choices, e.g. the ones listed in the response, to validate that "median" is a better choice given the large experience diversity.

---

> > > ### Author Response · Authors · 2021-07-14
> > > **Response to Reviewer Y1Wu**
> > >
> > > Thank you for your response. We acknowledge the presence of annotator bias in the introduction (L74) and view it as a research opportunity. We will consider adding more discussion specifically on the tradeoff between the number of examples and number of annotations per image. We had to make a decision regarding this tradeoff early in the planning phase, as we had only one chance to collect expert annotations.
> > >
> > > Having said that, we concur with you that the number of annotations per training image (which may increase the quality of the label) could rival the importance of the number of images. In fact, this defines a future direction for this project - we are making our annotation tool public and encouraging other experts in the field to contribute to the dataset by providing even more annotations for the mammograms in CSAW-M.
> > >
> > > Regarding including a comparison of median and other options as the GT, this represents a substantial amount of work that cannot be completed before the end of the discussion period. However, as you suggested, we will definitely consider adding a discussion/comparison of them in the camera-ready version of the manuscript.

---

> > > > ### Comment · Reviewer_Y1Wu · 2021-07-15
> > > > **Response to authors**
> > > >
> > > > Thanks to the authors for their further clarification. I am glad to know that they will make the annotation tool public and possibly include more expert annotations and I believe it will make the dataset stronger.
> > > >
> > > > Given the short notice of the discussion period, it is almost infeasible to include a thorough comparison of median and other solutions, but it would be nice to see a (even only compare to 1-2 solutions) comparison in the camera-ready version. But this is just a suggestion and would not be taken into account for the paper rating.

---

> ### Author Response · Authors · 2021-07-13
> **Response to Reviewer Y1Wu - Part 1**
>
> First, we would like to thank you for your thoughtful review and the valuable feedback you provided. By addressing your comments, we believe the quality of our paper has improved significantly. Below, we give a point-by-point summary:
>
> 1. Regarding your point about the hosting of CSAW-M: The SciLifeLab Data Repository is a data repository hosted by SciLifeLab (the Swedish National Initiative in molecular biosciences https://www.scilifelab.se/) for researchers to upload and share data. Currently, this service relies on Figshare (https://figshare.com/) and is maintained by the SciLifeLab Data Center. The Data Center plans to migrate to an in-house storage solution in the future, but this will not impact data availability in any way. CSAW-M will receive continued support from the Data Center (more information could be found in Appendix K, and also the support letter).
> \
> \
> To clarify, CSAW-M is currently not published, but we will publish it after the NeurIPS decision. The public train/test set may be requested by anyone who agrees to the terms of use (and signs up for a free Figshare account). The “planned service” refers to a planned infrastructure that allows researchers to upload trained models in a Docker container and evaluate them on the CSAW-M private test set using Kubernetes. This service is currently not supported by the Data Center, but will be offered in the future (see the support letter on page 28).
> \
> \
> Our description of the private test set evaluation service as a “controlled, less biased setting” was a bit unclear. We have clarified the text in the main paper (L59) which now reads:
>   > In the future, an evaluation service hosted at SciLifeLab will allow researchers to submit Dockers containing their models for evaluation on the private test set, as a control against overfitting to the public test set.
>
>   We have added a section to Appendix K giving further details (L635-L640).
>
>
> 2. It is true that CSAW-M is collected from a specific region. This limitation is discussed in the Broader Impact Statement (L331) where we consider possible sources of biases. However, we note that these types of local population biases are nearly unavoidable in medical datasets due to difficulties coordinating authorization for data sharing between administrative regions.
>
>
> 3. We did not list age information for other datasets in Table 1 because it is not readily available. The paper describing INBreast provides a distribution of age for different BI-RADS scores, but not the complete data. According to the paper, participants range from approximately 30-80 years. For MIAS and CBIS-DDSM, one can guess at the age distributions based on where the data was collected: e.g. according to Reference 1 (below) participants between 50-71 are offered breast screening in England, where MIAS was collected. Reference 2 mentions that in the USA participants 45-75 receive regular screening, though participants 40-44 can elect to have it as well, which may give clues as to the age range for CBIS-DDSM.
>
>
> 4. Based on your suggestion, we have updated the paper to provide details of the image preprocessing in Section 2 (L119-127)
>
>
> 5. Regarding the number of annotations for each training sample: the annotators were assigned an equal number of training images (~2,500) to annotate (L200), so the distribution among annotators is approximately equal.
> \
> \
> We agree that having more annotations per image is desirable and we considered this in the design phase. But any data collection process will have some limited annotation budget, and it is well-known that deep learning performance scales as data increases. Given a choice between adding 4x the number of examples or having more (4) annotators per training image, we believe the community benefits more by adding new examples. Although the individual training examples will contain more bias, the annotator agreement is generally strong, as indicated in Figure 5. Furthermore, our design choice creates an interesting opportunity to study annotator bias, as the test sets contain annotations from all 5 experts which can be compared against the training data.
>
>
> References:
>
> [1] https://www.nhs.uk/conditions/breast-cancer-screening/#:~:text=Breast%20screening%20is%20offered%20to,risk%20of%20developing%20breast%20cancer.
>
> [2]https://www.cancer.org/healthy/find-cancer-early/american-cancer-society-guidelines-for-the-early-detection-of-cancer.html

---

### Decision · Program_Chairs · 2021-07-27

**Decision:**

Accept

**Comment:**

The authors introduce a dataset mammographic dataset for benchmarking models in the domain of breast cancer. The reviewers noted the extensive scope of the dataset as well as its significance and there is a consensus on acceptance.